# Snow Depth Estimation on Slopes Using GPS-Interferometric Reflectometry

**DOI:** 10.3390/s19224994

**Published:** 2019-11-16

**Authors:** Haohan Wei, Xiufeng He, Yanming Feng, Shuanggen Jin, Fei Shen

**Affiliations:** 1College of Civil Engineering, Nanjing Forestry University, Nanjing 210037, China; 2School of Electrical Engineering and Computer Science, Queensland University of Technology, Brisbane 4001, Australia; y.feng@qut.edu.au; 3School of Earth Science and Engineering, Hohai University, Nanjing 210098, China; xfhe@hhu.edu.cn; 4Shanghai Astronomical Observatory, Chinese Academy of Sciences, Shanghai 200030, China; sgjin@shao.ac.cn; 5Key Laboratory of Virtual Geographic Environment (Nanjing Normal University), Ministry of Education, Nanjing 210046, China; shen.f@njnu.edu.cn

**Keywords:** GPS-IR, snow depth, multipath, slope

## Abstract

Snow is one of the most critical sources of freshwater, which influences the global water cycle and climate change. However, it is difficult to monitor global snow variations with high spatial–temporal resolution using traditional techniques due to their costly and labor-intensive nature. Nowadays, the Global Positioning System Interferometric Reflectometry (GPS-IR) technique can measure the average snow depth around a GPS antenna using its signal-to-noise ratio (SNR) data. Previous studies focused on the use of GPS data at sites located in flat areas or on very gentle slopes. In this contribution, we propose a strategy called the Tilted Surface Strategy (TSS), which uses the SNR data reflected only from the flat quadrants to estimate the snow depth instead of the conventional strategy, which employs all the SNR data reflected from the whole area around a GPS antenna. Three geodetic GPS sites from the Plate Boundary Observatory (PBO) project were chosen in this experimental study, of which GPS sites p683 and p101 were located on slopes with their gradients up to 18% and the site p025 was located on a flat area. Comparing the snow depths derived with the GPS-IR TSS method with the snow depth results provided with the GPS-PBO, i.e., GPS-IR with the conventional strategy, the Snowpack Telemetry (SNOTEL) network measurements and gridded Snow Data Assimilation System (SNODAS) estimates, it was found that the snow depths derived with the four methods had a good agreement, but the snow depth time series with the GPS-IR TSS method were closer to the SNOTEL measurements and the SNODAS estimates than those with GPS-PBO method. Similar observations were also obtained from the cumulative snowfall time series. Results generally indicated that for those GPS sites located on slopes, the TSS strategy works better.

## 1. Introduction

Snowpack is one of the most critical sources of freshwater, which influences the global water cycle and climate change [1]. Over one sixth of the world’s population relies on water runoff from seasonal snow packs and glacier melt [2]. By measuring the snow depth, we can estimate the amount of fresh water stored in snow. Traditionally, measuring snow depth is done manually or with some automated ground-based sensors [3]. The snow depth measurements with these techniques are accurate, but costly or suffer from low temporal and spatial resolution. Satellite optical remote sensing techniques have been used to measure snow depth at a moderate resolution, but cloud cover always makes their operation difficult [4,5]. Airborne and ground-based LiDAR are able to accurately estimate snow depth over large areas, but they are expensive, difficult to be automatic, and need precise terrain elevations [6]. Nowadays, in addition to the traditional applications [7,8,9], GPS reflectometry (GPS-R) methods have the potential to detect snow depth with high temporal and spatial resolution based on their physical reflection properties and polarization characteristics of GPS signals reflected from ground surface [10,11]. For GPS-R applications, GPS receivers should be installed with two unique antennas to receive direct signals and reflected signals separately [12]. Recently, scientists found that land surface parameters, including soil moisture [13,14], snow depth [6,15,16], and vegetation water content [17], can be derived from a single geodetic GPS receiver. This technique, known as GPS Interferometric Reflectometry (GPS-IR), utilizes multipath signal-to-noise ratio (SNR) data relating to the changes of ground surface parameters around a GPS receiver [18]. With the change in antenna heights, the frequency of SNR time series caused by direct and reflected signals is changing, and the snow depth can be derived from the differences between reflected antenna heights.

In the past 30 years, thousands of permanent operating geodetic GPS receivers have been installed around the world, which provide data to derive land surface parameters in different conditions using the GPS-IR method. Bilich [19] showed that the variability of SNR data due to multipath has a strong dependence on the changes in the antenna environments. As for snow depth estimation, Larson [15,18] indicated that there is a good agreement between the snow depth derived with traditional geodetic GPS receivers and in situ ultrasonic sensor measurements of the Snowpack Telemetry (SNOTEL) network. Nievinski [20,21] presented forward/inverse models for snow depth estimation with the GPS-IR. A new approach was developed for snow depth estimation based on multipath reflectometry and geometry-free triple-frequency signals using a geodetic GPS receiver [22]. The SNR data from GPS L2C signal, as well as L1C/A, L2P, and L5 signals, are proposed to derive the snow depth, the results show that the SNR from GPS L2C signal is preferable or comparable to those from the other signals [16,23,24,25]. Boniface et al. [26] made a comparison between the gridded snow depth estimates from the Snow Data Assimilation System (SNODAS) and the snow depth observations derived from the GPS-IR in the western United States. The results showed that at 80% of the sites there was a good agreement between the data derived with these two methods, while significant differences existed at the sites located in a complex terrain or in the areas with strong vegetation heterogeneities. Previous studies of the snow depth derived with the GPS-IR method show that good results are obtained at the GPS sites located in flat areas, while topographical environments have significant effects on the results. The direct signals are sent from GPS satellites and are reflected by land surfaces from different azimuths and elevation angles; as a consequence, the reflected signals received by GPS receivers will be unavoidably affected by topographic inequality. This limits the potential of using large number of GNSS sensors deployed on mountainous areas and hill landforms to collect snow depth data. However, snow depth data from mountainous areas are of more significance.

This paper analyzes the influences of topographical characters on snow depth estimation with the GPS-IR method. A strategy called the Tilted Surface Strategy (TSS) is proposed for those GPS sites located on slopes to reduce the topographic influences. Finally, the GPS-IR snow depths derived with the TSS method, as well as those using the conventional strategy, are compared and validated with the snow depth measurements with in situ sensors of the SNOTEL network and gridded SNODAS estimates. This study is based on a data set collected in the western USA, and therefore the data source is first described in the paper.

## 2. GPS Observations and Snow Depth Measurements

### 2.1. GPS Observations

The Plate Boundary Observatory (PBO) network [27] of the EarthScope is mainly used for the determination of the boundary plate deformations in the Western United Sates, and has hundreds of GPS stations. Many of the PBO GPS sites are located in hill landforms. Consequently, multipath errors at these sites are primarily caused by ground surface reflection, such as snow pack, soil surface, and vegetation. This provides an opportunity to study the changes of snow depths derived with the GPS-IR method in slopes.

In order to demonstrate how to estimate and validate the snow depth on a slope using the GPS data, three GPS sites of PBO were selected (see Figure 1 and Table 1). The PBO site p025 is located on a flat ground surface in suburban area of Montana, p101 is located in a mountain area in the Crawford Mountains of Utah, and p683 is located in a mountain area in the Chesterfield Range of Idaho. Both p101 and p683 are located on slopes with a surface slope up to 18% (Figure 2). The three sites were all equipped with Trimble NetRS receivers and antennas (model of TRM29659 with SCIT radome). The receiver phase centers were 1.9 m above the ground. All GPS data were collected at a rate of 15 s, and covered a full water-year 2015, from October 2015 to April 2016, allowing for a complete process from snow fall to snow melt.

### 2.2. SNOTEL Measurements

The Snowpack Telemetry (SNOTEL) network [28] provides in situ snow depth measurements in the western United States using ultrasonic sensors. The daily snow depths are provided by the National Water and Climate Center (NWCC) [29]. In order to validate the snow depths estimated from GPS observations, three SNOTEL sites (ID: 1053, 374, 770) near the selected PBO GPS sites were chosen. All SNOTEL sites were located on flat and vegetation-free surfaces. The information is summarized in Table 2. The distances between the GPS sites and corresponding SNOTEL sites were 14 km, 15 km, and 29 km, respectively, while the elevation differences were 376.1 m, 417.9 m, and 6.2 m, respectively. Due to the large height differences and long distances between GPS sites and SNOTEL sites, gridded SNODAS snow depth estimates were also used to evaluate the GPS-IR derived snow depths with the TSS method.

### 2.3. SNODAS Gridded Products

SNODAS is a modeling and data assimilation system [30] developed by the National Snow and Ice Data Center (NSIDC) for the continental United States. By integrating the snow data from airborne and satellite platforms, and ground stations with output from Numerical Weather Prediction (NWP) models [31,32], SNODAS provides a physically-based, energy-and-mass-balance, spatially-distributed, multi-layer, and mass-balance snow model. This snow model is run with daily output at 30-arc-second resolution (about 1 km). Gridded SNODAS data were obtained from NSIDC for the 2015 water year. The data were processed using a MATLAB software package to extract snow depths for grid cells. It should be noticed that gridded SNODAS snow depth estimates are point values for the center of each grid cell, instead of an area value [23]. In order to align with the locations of each GPS site, the SNODAS gridded snow depth estimates were interpolated at each GPS site using a bilinear interpolation [26].

## 3. Methodology

### 3.1. GPS-IR Snow Depth Estimation in Flat Areas

In typical applications of GPS positioning, the multipath effect is regarded as one of the major error sources. The reflected and direct signals were collected by a GPS receiver at the same time, and the signal-to-noise ratio (SNR) could be calculated, which was used as an indication of the signal quality. Due to its characteristics, the SNR was sensitive to the interference between direct or line-of-sight power (*P_d_*) and reflected power (*P_r_*) [16]:(1)$$SNR\propto P_{d}+P_{R}+\sqrt{P_{d}P_{r}}cos\phi ,$$
where ϕ is the reflection phase. The SNR is a function of signal wavelength, elevation angle of a satellite, and the GPS antenna height above the reflecting surface. It is mainly determined by the antenna gain pattern and by the satellite transmitted power [16]. Previous studies showed that the effects of antenna gain pattern dictate that *P_r_* << *P_d_*, which means the multipath contribution to SNR is small in magnitude/power but oscillatory, while *P_d_* is large in magnitude/power but goes through a complete cycle only once over a satellite pass (Figure 3a) [13]. For GPS multipath reflectometry applications, the trends in the SNR are not of interest and can be removed. After the trend power (*P_d_*, *P_r_*) is removed using a low-order polynomial, the simplified SNR multipath pattern can be described as [16]:(2)$$SNR_{d}=A\cos\left(4\pi h\lambda^{-1}\sin\left(e\right)+\varphi \right),$$
where the parameter *A* represents the amplitude of the detrended SNR, *h* is the reflected height, which is the vertical distance between the antenna phase center and the reflecting surface, *λ* is the GPS carrier wavelength, *e* is the elevation angle of a GPS satellite, and φ is the phase shift. According to previous studies [25,33], the signal power quality of L2C carrier frequency is superior to others and it was more suitable for GPS snow depth determination in this case. As an example, the original SNR curves and detrended SNR curves of PBO GPS site p025 are shown in Figure 3a,b, respectively.

The detrended SNR can be modeled with Equation (2). For a given parameter *h*, the detrended SNR curve should have a constant amplitude and frequency, which is a function of elevation angle *e*. For snow depth determination, the reflected height *h* was unknown and could be derived using the Lomb–Scargle periodogram (LSP) method [34]. The LSP gives a series of frequencies of the interferogram, and the dominant frequency can be converted to the reflected height using the following equation [35]:
(3)$$H_{r}=\frac{1}{2}\lambda f,$$
where *H_r_* is the reflected height of GPS antenna converted from dominant frequency, *f* is the dominant frequency derived from LSP, *λ* is GPS carrier wavelength (for L2 carrier, *λ* = 0.244 m). It should be noted that the reflected height of GPS antenna was not the real antenna height, i.e., the receiver phase was centered above the ground. In practice, the reflected height varies with satellite tracks and is affected by several factors, e.g., surface roughness, satellite elevation cut-off angle, orientation and tilt angle of the reflecting surface, and the cover conditions (e.g., snow, vegetation) of the reflecting surface. Therefore, the snow depth could be derived by differencing the two reflected heights between the bare ground and the snow-covered ground.

Figure 3 shows the LSP curves from detrended SNR data with elevation cut-off angles of 5°–20° [16,25] at site p025. The black solid line refers to bare ground on day of year (DOY) 2015-312 and the blue dot-and-dash line and red dash line refer to snow-covered ground on DOY 2015-333 and DOY 2016-019, respectively. It also indicates that with the accumulation of snowfall, the powers and amplitudes of LSP curves decreased, and the main reflected heights became smaller. Thus, the snow depths derived with the GPS-IR method can be calculated as the differences between the corresponding reflected heights. The reflected heights were 2.20 m, 2.07 m, and 1.95 m using Equation (3), and therefore the snow depths were 0 m, 0.13 m, 0.25 m, respectively. Compared with the snow depths of 0 m, 0.08 m, and 0.33 m, recorded by in situ sensors near the site p025, the snow depths determined with the GPS-IR method are quite accurate.

### 3.2. GPS-IR Snow Depth Estimation on Slopes

For GPS multipath applications and studies, it is helpful to display the distribution of multipath reflected points around the GPS antenna. Figure 4 shows the specular reflected point tracks of L2C signals for eight GPS satellites tracked at site p025. As previous studies indicated [13], reflected point tracks extended from 5° to 20° are used to estimate the powers and frequencies of the multipath modulations. It should be noted that the specular reflected point tracks were obtained in an ideal case, while the actual reflected areas can be presented by means of GPS footprint derived from the First Fresnel Zone (FFZ) [36].

In order to illustrate the differences of FFZ footprints located on a flat surface and a slope, Figure 5 shows the footprints of FFZ for satellite 7 and satellite 8 on DOY 2015-312 at site p025 on a flat surface (Figure 5a), and footprints of FFZ for satellite 26 on DOY 2015-356 at site p101 on a slope (Figure 5b). It can be seen from Figure 5a that for a flat reflecting surface, each single FFZ footprint spread and covered up to about 50 m. As for a slope surface, as shown in Figure 5b, where the surface slopes were 14% (northeast), 0% (northwest), 7% (southeast), and 18% (southwest), the footprints of FFZ in the northwest quadrant in flat area showed a normal level, as in Figure 5a, but the ranges of footprints on the others three quadrants were shrunken with the slope angles increasing. This is because all the slopes were going down from the antenna. With the incident angle getting higher, the ranges of FFZ footprints will shrink, especially for the northwest quadrant with a maximum slope angle of 18%. As the footprint ranges shrink, the power of the reflected signals received by the antenna will be obviously weakened. It is also shown in Figure 6b that for reflected signals from slopes on the northeast (brown line), southeast (black line), and southwest (pink line), which were going down away from the antenna with slope angles of 14%, 7%, and 18%, respectively, the signal power weakened. On the other hand, if the slopes went up from the antenna, the ranges of FFZ footprints would be greatly stretched, even some of the reflected signal would be missing if the satellite’s elevation angle was smaller than the surface’s slope. In general, both upslope and downslope from the antenna, with the slope angle increasing, the power of the reflected signal would be weakened. 

Figure 6a shows the distribution of GPS signal footprints of FFZ at site p101 in different directions. The footprints with brown, blue, black, and pink are for the directions of northeast, northwest, southwest, and southeast, respectively, which had the surface slopes of 14%, 0%, 7%, and 18%, respectively. Figure 6b shows their LSP curves derived from detrended SNR data. It shows that the SNR amplitude for a flat area was significantly stronger than those for tilted areas. With slope increasing, the amplitude of the LSP curve became smaller, and even unrecognizable when the slope reached 18%. Furthermore, Figure 7 shows the original SNR time series, detrended SNR time series, and reflected heights from LSP curves at PBO GPS sites p101 (left) and p683 (right) on a flat area (northwest direction) and tilted area (southeast direction), respectively. We can also see that the oscillations of the SNR time series on flat areas were more obvious than those on tilted areas, and especially the amplitudes of LSP curves on flat areas were larger than those on tilted areas. It is clear that for GPS sites located on a slope, the reflected signals will be scattered and the power will be greatly reduced due to the angles and orientations of tilted areas. For the snow depth determination with the GPS-IR, it is necessary to find a flat area around the GPS site located on the tilted ground surface.

For the GPS sites on slopes, the Tilted Surface Strategy (TSS) that uses SNR data reflected on flat quadrants was used to derive the snow depth instead of the conventional strategy, which uses all the reflected SNR data around the antenna. With reference to the methods provided by Larson [16], the TSS for daily snow depth estimation is summarized as follows:
(1)Reflected point tracks located on a flat area as well as with strong ground reflection should be chosen using the LSP method;(2)Tracks with the elevation angles from 5° to 20° are chosen. LSP curves with dominant peaks smaller than three times their background noise are removed;(3)Reflected height of each selected track can be calculated from Equation (3). Mean daily reflected height as well as standard deviation can be determined from all available reflected heights;(4)Snow-free reflected heights are estimated using summertime data, and snow-covered reflected heights are estimated using snow-time data. Therefore, the daily mean snow depths are calculated by the difference between snow-free and snow-covered reflected heights.

In general, the main difference to the conventional strategy which GPS-PBO utilized is that the TSS identifies the reflected point tracks located on a flat area or near flat directions while removing the non-flat areas, and then selects the highest amplitudes in the LSP curves to determine the reflected heights.

## 4. Results and Analysis

### 4.1. Snow Depth Time Series

Following the TSS method presented in Section 3, the snow depths were derived from the SNR data of the GPS sites p683 and p101, which were both located on slopes with surface slopes up to 18% (see Figure 2). Moreover, the snow depths provided by the PBO project and in situ measurements of SNOTEL as well as SNODAS estimates were used for the purpose of comparison and validation. Figure 8 shows the snow depth time series at three selected GPS sites during water year 2015. These time series were produced with all the methods: GPS-PBO (Figure 8a–c), SNOTEL (Figure 8a–c), SNODAS (Figure 8a–c), and GPS-TSS (Figure 8b–c). As one can see, they are in general agreement in the overall seasonal pattern of snow accumulation and melt. Individual snowfall events can also be seen in the time series.

At p025 on a flat surface, Figure 8a shows a good consistency in terms of the fluctuation and trend of the snow depth time series produced by three methods: GPS-PBO, SNOTEL, and gridded SNODAS, although there was an obvious elevation difference of 376.1 m and great separation of 14 km between GPS site p025 and SNOTEL site 1053 (see Table 2). This suggests the GPS-IR in the conventional strategy (GPS-PBO) works well on flat areas. Moreover, the time series between GPS-PBO and gridded SNODAS estimates show a better agreement than those between GPS-PBO and SNOTEL measurements. This may be due to the spatial distance between the GPS site and SNOTEL site. Through a bilinear interpolation method, gridded SNODAS estimates were in good agreement with the GPS-PBO results. The correlation coefficients between the GPS-PBO and SNOTEL, and GPS-PBO and SNODAS were 0.9266, 0.9647, respectively, with standard deviations being 0.0877 m and 0.0138 m, respectively (Table 3), which again shows that the snow depths derived with GPS-PBO, SNOTEL, and SNODAS were consistent, and the results with SNODAS were closer to the GPS-IR results.

In order to evaluate the proposed TSS method, the snow depths derived with the GPS-TSS method are shown for p683 and p101 as well as other three methods (Figure 8b,c). Despite the general agreements among the results with these four methods, the snow depth time series of SNOTEL were always greater than the others, especially for site p101, where the elevations of the SNOTEL in situ sensor and GPS site were 2434.0 m and 2016.1 m, respectively. This may be due to the higher elevations of the two SNOTEL sites and the longer distances between SNOTEL sites and GPS sites. A better agreement can be seen among the results with GPS-PBO, GPS-TSS, and gridded SNODAS. Furthermore, the time series of the GPS-TSS results were closer to SNODAS estimates than the GPS-PBO results. In addition, the difference between GPS-PBO and GPS-TSS was much smaller than the difference to SNODAS; this is mainly because these two GPS methods used the same data set, i.e., GPS-TSS utilized some of the SNR data, while GPS-PBO used all of the SNR data. This again proves the reliability of GPS-TSS as well as its better accuracy than the GPS-PBO method.

### 4.2. Cumulative Snowfall

Cumulative snow depth computed for a water year is of particular interest for water resource availability and water budget characterization [26]. The annual cumulative snowfall (ACSF) of a water year can be computed as follows,
(4)$$ACSF=\sum_{d=1}^{365}\left(\Delta +\right)SF_{d}$$
where *d* is the day index of a water year, *SF* denotes Snowfall for a single day, Δ+ denotes positive increments in snow depth, for a no-snowfall day or snow-melt day, Δ+ is zero.

The ACSF time series for the three GPS sites p025 (on flat surface), p683 (on slope), and p101 (on slope) were computed with GPS-PBO, GPS-TSS, SNOTEL, and SNODAS, and are plotted in Figure 9. As can be seen, they agreed well until early February (p025) and early April (p683, p101). In particular, although the derived quantities of cumulative snowfall were different, snowfall timings generally agreed among these methods. Most individual snowfall events were clearly recorded by these methods. In summary, for p683 (Figure 9b) and p101 (Figure 9c) on slopes, the GPS-TSS time series were always in between the time series of GPS-PBO and SNODAS, and slightly closer to SNODAS curves than GPS-PBO curves.

## 5. Discussion

Although GPS multipath signals are considered error sources in traditional applications, GPS-IR provides a method to detect snow depth with high temporal and spatial resolution without any additional equipment. In order to improve the accuracy and reliability of snow depth derivation for GPS sites located on slopes, we proposed the TSS strategy, which uses the SNR data reflected only from the flat quadrants.

As described in Section 3.2, the GPS-TSS method only utilizes SNR data reflected from the flat quadrants, while GPS-PBO, which is considered the conventional strategy, uses all of the SNR data surrounding the site [16]. This means both GPS-TSS and GPS-PBO had the same data source. Figure 10 shows the comparison of snow depth estimations between GPS-PBO and GPS-TSS at p683 and p101, which were located on slopes. The correlation coefficients were 0.98 and 0.99, the mean deviations were −0.111 m and 0.019 m, and the standard deviations were 0.085 m and 0.048, respectively. As the feasibility of the conventional strategy GPS-PBO has already been proven by previous studies [6,11,16,19], this comparison again proved that the GPS-TSS can be used efficiently and reliably to derive snow depth.

Table 4 and Table 5 show the snow depth comparisons between GPS-TSS, GPS-PBO and SNOTEL, SNODAS for sites p683 and p101, located on slopes, and list the correlation coefficients, standard deviations, and mean deviations between them. All these indicate that for GPS sites located on slopes, the GPS-TSS method was slightly better than the conventional strategy method of GPS-PBO. Moreover, the correlation coefficients in Table 5 are lower than those in Table 4, which is mainly due to the larger spatial distance between GPS sites and in situ sensors. As listed in Table 2, the elevation differences between GPS sites and SNOTEL in situ sensors of p683 and p101 were 6.2 m and 417.9 m, respectively. The horizontal distances between GPS sites and SNOTEL in situ sensors of p683 and p101 were 29 km and 15 km, respectively. Furthermore, the SNOTEL network is one of the major data sources assimilated into SNODAS. The larger spatial distance leads to the lower correlation coefficients.

## 6. Conclusions

It was proven that the GPS-IR method can produce good results of snow depth from GPS SNR data. However, the snow depth estimates were affected by the topographic conditions around a GSP site due to scattering of the reflected signals. As the reflected footprints are always around the GPS sites, we found that the powers of the signals reflected from slope areas are diffused and weakened, while the reflected signals from flat areas show a normal level. This will inevitably affect the snow depth results derived with GPS-IR conventional strategy, which employs all reflected signals around the GPS sites located on slopes. In this paper, GPS-IR snow depths derived on slopes were considered and the Tilted Surface Strategy was proposed. Comparing with the snow depths from SNOTEL measurements and gridded SNODAS estimates at selected GPS sites p683 and p101 for water year 2015, we demonstrated the snow depths derived with the four methods had a general agreement, but the snow depth time series of GPS-IR with TSS methods were closer to the SNOTEL measurements and SNODAS estimates than those with GPS-IR conventional methods. That is, the correlation coefficients were in the same level, while both standard deviations and mean deviations showed that the results of GPS-TSS were always better than results of GPS-PBO (Table 4 and Table 5). Similar results can also be seen in the cumulative snowfall time series. Both the snow depth time series and cumulative snowfall time series proved that GPS-TSS can monitor both snow accumulation and snowmelt better. They also revealed that for the snow depth derivation using the GPS-IR method, the Tilted Surface Strategy can improve the results for the GPS sites located on slopes.

Besides the tilted ground surfaces, the surface roughness, vegetation coverage, as well as wind redistribution and radiation balance should also be considered for GPS-IR snow depth derivation. Furthermore, with the development of Global Navigation Satellite Systems (GNSS) like GLONASS, Galileo, and BeiDou, there will be more data for the study and application of GNSS-IR in the future. As the SNR measurements are available from low-end GNSS receivers, the deployment of GNSS sensors in mountain areas could be a low-cost solution for wide-scale snow depth data collections. With the TSS method, the snow depth from the GNSS sensors in mountain areas can still achieve good accuracy.

## Figures and Tables

**Figure 1 sensors-19-04994-f001:**
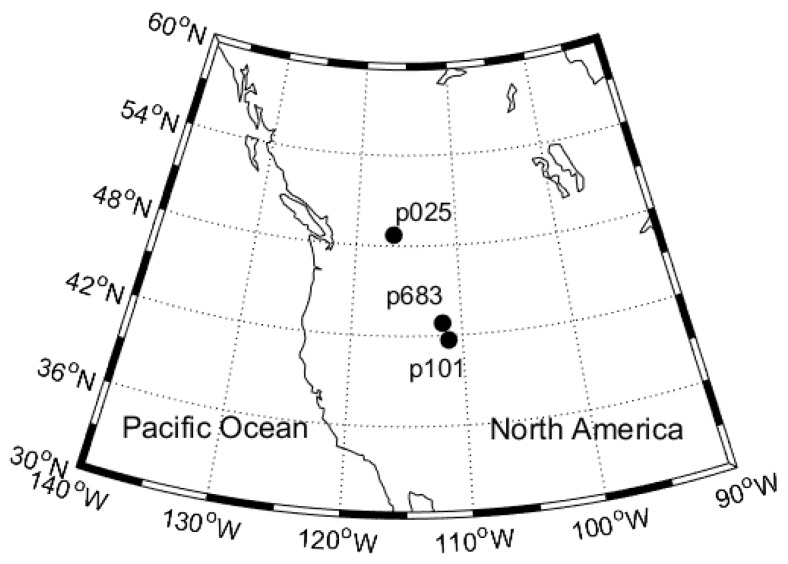
Three selected Global Positioning System (GPS) sites of the Plate Boundary Observatory (PBO) network in the western United States.

**Figure 2 sensors-19-04994-f002:**
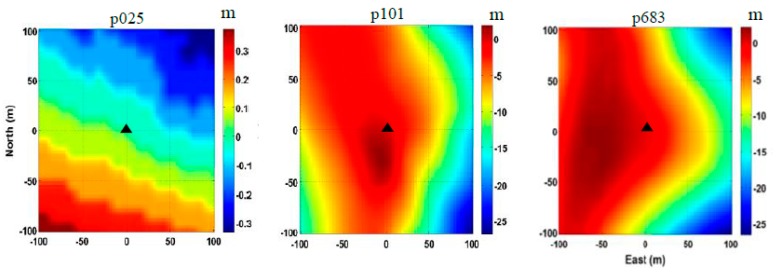
Digital elevation models (DEMs) for the selected GPS sites of p025, p101, and p683, black triangles denoting the location of GPS sites (the value at the center of DEM diagram is 0 m [27]).

**Figure 3 sensors-19-04994-f003:**
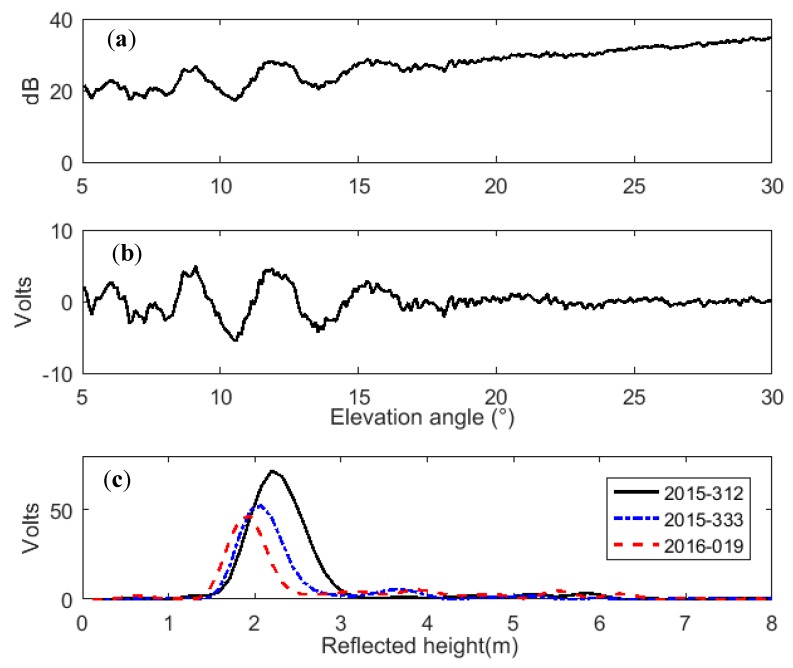
Recorded L2C signal-to-noise ratio (SNR) data for satellite PRN7 at p025. (**a**) Original L2C SNR data on day of year (DOY) 321, 2015; (**b**) detrended SNR data with the trend removed by a second-order polynomial; (**c**) reflected height from the detrended SNR data with the elevation angle ranging from 5° to 20° by Lomb–Scargle periodogram on DOY 2015-312, 2015-333, and 2016-019.

**Figure 4 sensors-19-04994-f004:**
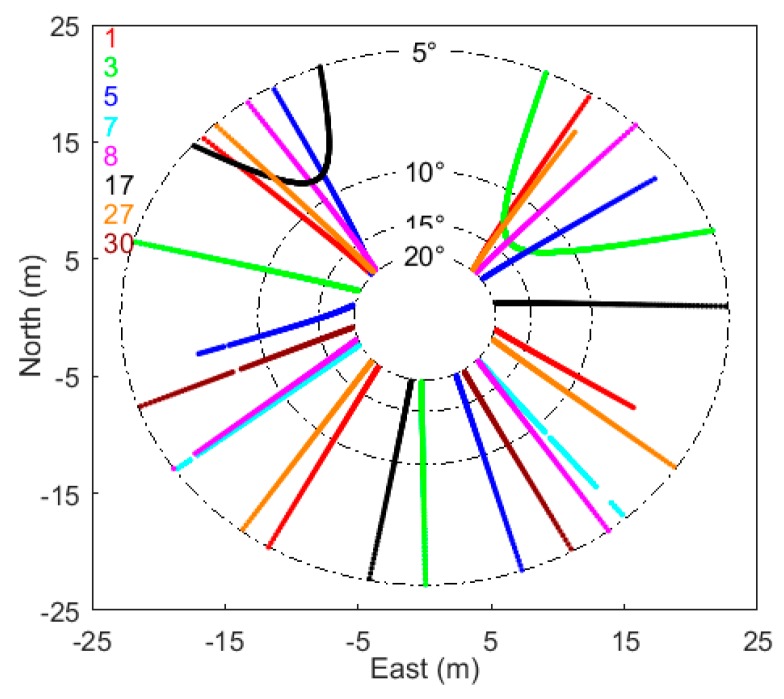
Specular GPS multipath reflected point tracks at p025 (DOY 2015-312). Red, green, blue, cyan, purple, black, orange, and brown lines represent specular reflected point tracks of GPS L2C signals from satellites 1, 3, 5, 7, 8, 17, 27, 30, respectively.

**Figure 5 sensors-19-04994-f005:**
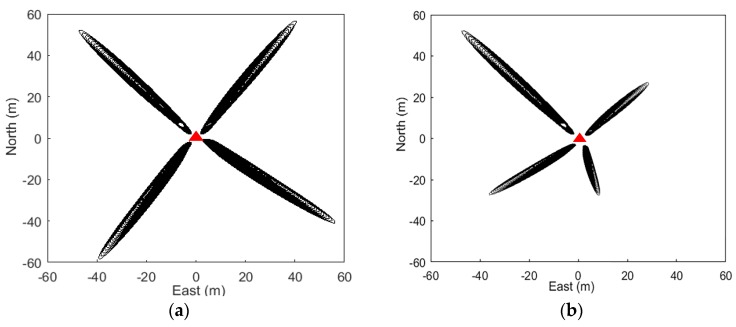
(**a**) First Fresnel Zone (FFZ) of satellite 7 (southwest and southeast) and satellite 8 (northwest and northeast) at p025 on a flat surface on DOY 2015-312; (**b**) FFZ of satellite 26 at p101 on a slope on DOY 2015-356, with the surface slopes of 14% (northeast), 0% (northwest), 7% (southeast), and 18% (southwest). All slopes went down from the antenna. Locations of GPS sites are shown with red triangles.

**Figure 6 sensors-19-04994-f006:**
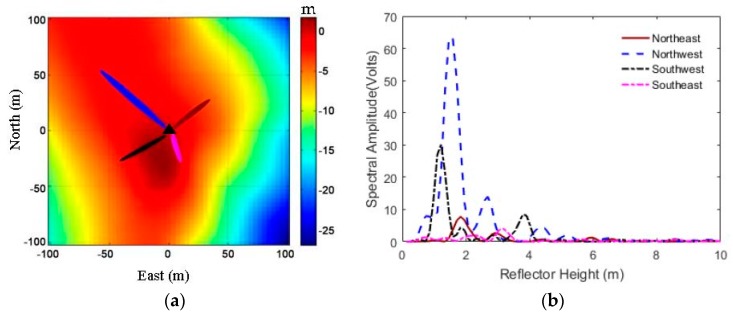
GPS footprints and Lomb–Scargle periodogram (LSP) curves for satellite 8 at GPS site p101 on DOY 2015-356. (**a**) Distribution of the GPS signal footprints of FFZ in different directions, with GPS site location shown in a black triangle; (**b**) LSP curves of each SNR data. Brown, blue, black, and pink lines represent GPS reflected signals in the directions of northeast, northwest, southwest, and southeast, respectively.

**Figure 7 sensors-19-04994-f007:**
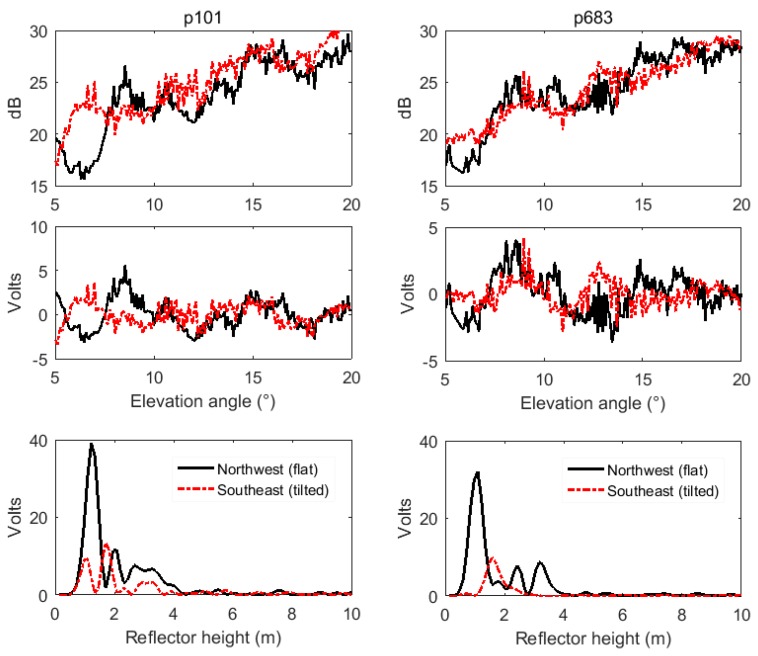
Original and detrended SNR time series and reflected heights by LSP on the same site with different directions. Left: PBO site p101 of SNR data from satellite 26 in the Northwest quadrant (flat area) and Southeast quadrant (tilted area) on DOY 2016-032; Right: PBO site p683 of SNR data from satellite 15 in the Northwest (flat area) and Southeast quadrants (tilted area) on DOY 2016-010.

**Figure 8 sensors-19-04994-f008:**
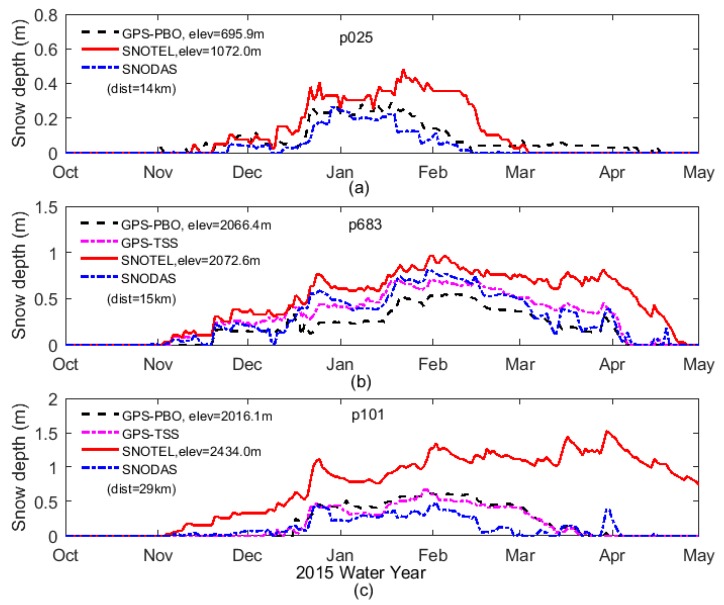
Snow depth time series derived with the GPS-PBO, GPS-TSS, SNOTEL, and Snow Data Assimilation System (SNODAS) during water year 2015 for p025 (**a**), p683 (**b**), and p101 (**c**).

**Figure 9 sensors-19-04994-f009:**
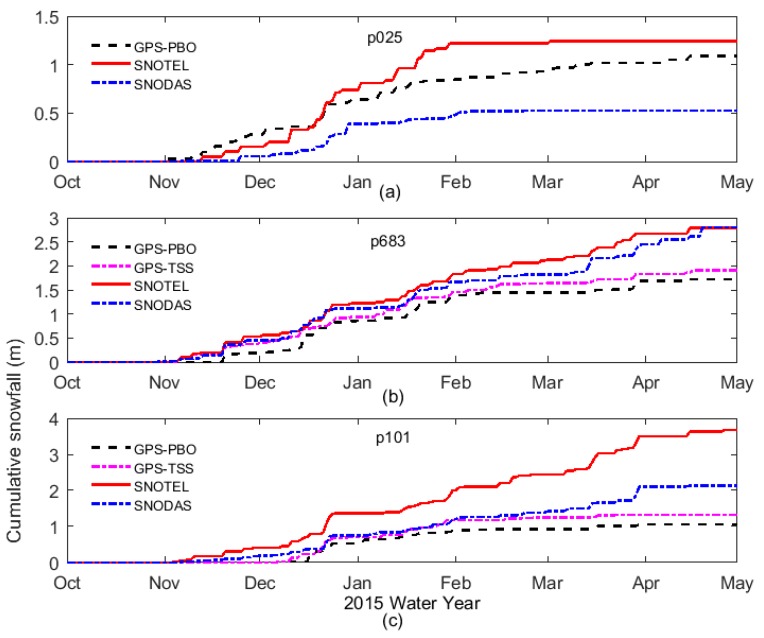
Time series of cumulative snowfall computed with GPS-PBO, GPS-TSS, SNOTEL, and SNODAS during water year 2015 for p025 (**a**), p683 (**b**), and p101 (**c**).

**Figure 10 sensors-19-04994-f010:**
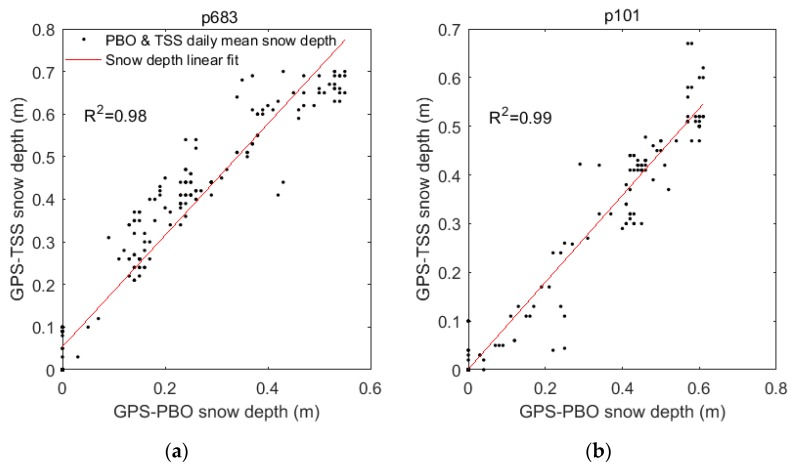
Comparison of snow depth estimations derived with GPS-PBO and GPS-TSS at p683 (**a**) and p101 (**b**). Black dots and red lines represent the daily mean snow depth derived from the GPS-PBO method and GPS-TSS method, and snow depth linear fit, respectively.

**Table 1 sensors-19-04994-t001:** Locations of the selected GPS sites.

PBO Site	Latitude Degrees	Longitude Degrees	Elevation (m)	Surface Condition
p025	48.731	−116.288	695.9	Flat
p101	41.6923	−111.2360	2016.1	Tilted
p683	42.8267	−111.7345	2066.4	Tilted

**Table 2 sensors-19-04994-t002:** Locations of in situ Snowpack Telemetry (SNOTEL) sites.

Site ID	Corresponding PBO Site	Latitude Degrees	Longitude Degrees	Elevation (m)	Elevation Difference to PBO Site(m)	Horizontal Distance to PBO Site (km)
1053	p025	48.723	−116.463	1072	376.1	14
374	p101	41.69	−111.42	2434	417.9	15
770	p683	42.95	−111.36	2072.6	6.2	29

**Table 3 sensors-19-04994-t003:** Comparisons of snow depth estimates with different methods at p025 (on flat surface).

Method	Correlation Coefficient	Standard Deviation (m)	Mean Deviation (m)
PBO vs. SNOTEL	0.9266	0.0877	−0.0420
PBO vs. SNODAS	0.9647	0.0138	0.0307

**Table 4 sensors-19-04994-t004:** Comparisons of snow depth estimates with different methods at p683 (on tilted surface).

Method	Correlation Coefficient	Standard Deviation (m)	Mean Deviation (m)
PBO vs. SNOTEL	0.9250	0.1850	−0.2696
PBO vs. SNODAS	0.9706	0.1066	−0.0880
TSS vs. SNOTEL	0.9521	0.1382	−0.1583
TSS vs. SNODAS	0.9704	0.0859	0.0233

**Table 5 sensors-19-04994-t005:** Comparisons of snow depth estimates with different methods at p101 (on tilted surface).

Method	Correlation Coefficient	Standard Deviation (m)	Mean Deviation (m)
PBO vs. SNOTEL	0.7290	0.4014	−0.5365
PBO vs. SNODAS	0.8927	0.1480	0.0781
TSS vs. SNOTEL	0.7274	0.4034	−0.5558
TSS vs. SNODAS	0.8902	0.0132	0.0588

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
