# Peer review of "Snow Depth Estimation on Slopes Using GPS-Interferometric Reflectometry"

_sensors, 2019, doi:10.3390/s19224994_

Round 1

Reviewer 1 Report

The authors propose to only consider reflections on flat areas around the antenna to improve snow depth measurements from GPS interferometry.  The subject is interesting and the overall paper quality is good. 

However, the idea of excluding quadrants is not new (for example it is applied in M. Durand and A. Rivera, "GPS reflectometry study detecting snow height changes in the Southern Patagonia Icefield", Cold Regions Science and Technology, July 2019), and the contribution of this paper is relatively small in my opinion.  The comparison between the standard GPS-PBO and the author's TTS technique is too limited (only 2 sites during a single water year and with no in-situ independent measurements) to draw a clear conclusion on the advantage of the technique.

Some specific comments:

Figure 2: indicate the orientation of the photographs (where is North?).

section 3.2: clarify the sign of the slope.  I assume that the antenna is on a local hill, but this is not clear from the photographs in Fig 2.  It needs to be confirmed and made clear in the text.   

line 221: "will be greatly stretched" : this should be further explained as it is counter-intuitive to me.  I would expect the FFZ to shrink as the incident angle is higher on a tilted surface (assuming again that the terrain goes down away from the antenna).  A drawing illustrating the tilted surface and the position of the reflection point in the southeast quadrant would help the reader.

line 222: "will be much weakened": this also deserves some explanation.  What is the influence of the antenna gain pattern, especially for the negative incident angles of the reflections?

Figure 5: explain why there are 4 orthogonal FFZ.  What do they correspond to?  In Figure 4, satellite 7 had only two reflection tracks.  Why the difference?

Figure 10: explain what the dots represent (average daily values?)

Some typos:

l78: the sentence starting with "The direct signals" is grammatically incorrect.

l81: "of making using" should be "of using"

l143: remove "subsection"

l158: use same "phi" symbol as in Eq (2)

l236: "the slope up to 18%" should be "the slope reaches up to 18%"

l266: "with removing" should be "while removing"

l271: "plat" should be "flat"

l305: "there shows" should be deleted

l347: "which located" should be "which are located"

l348: "with the standard deviations are": "are" should be "being"

Reviewer 2 Report

In this manuscript, authors proposed a strategy called the Tilted‐Surface‐Strategy (TSS) that uses the SNR data reflected only from the flat quadrants to estimate the snow depth instead of the conventional strategy which employs all the SNR data reflected from the whole area around a GPS antenna.

Major suggestions:

In section 4.2, based on the definition of the annual cumulative snowfall (ACFS), so snowfall is assumed to be a positive always increasing quantity? Please justify the wind, for example, it can displace snow. In Section 4.1 and 4.2, you analyzed the site p025, p683 and p101. But in Section 5, you just gave the analysis about p683 and p101. Why didn’t analyze the flat site p025? Tables 4 and 5 show the correlation coefficient at site p683 and p101. Could you explain why the correlation coefficients in Table 5 are lower than in Table 4?

Minor suggestions:

All Tables should be careful with the number of significant figures used for expressing the results. Table 1: Lat. (deg.), Lon. (deg.), Elevation (m). You should write the complete word before using some abbreviations. Check other Tables. P347, Tables 4 and 5, the correlation coefficients should be careful with the number of significant figures used for expressing the results. In Table 1, p683: Tilted. Check other Tables. The units, like meters (m), should be used with a space. Equation or figure should be with a capital E or F. Some descriptions, like P025 or p025, should be used in the same way in the whole manuscript. Some descriptions, like the Tilted‐Surface‐Strategy (TSS), should be used in the first time in the Abstract and Introduction. Then, just use TSS is fine. GPS‐TSS and GPS‐TTS, TSS and TTS, are they the same description? They should be checked in the whole manuscript. Some descriptions, like Figure 8a‐c with more than one figure, should use Figures. Conclusions: … can improve the results a bit …. Need to quantify.

Round 2

Reviewer 1 Report

The authors addressed some of the comments from the first review.  However,  the following comments still need to be addressed in my opinion:

1. Old comment, Figure 2: indicate the orientation of the photographs (where is North?)

The authors added a North indication to the elevation models, but it is still unclear whether the photographs are also oriented looking towards North.  This should be clarified to avoid incorrect interpretation, as the photographs do not seem to directly correspond to the DEMs.

2. Old comment, section 3.2: clarify the sign of the slope.  The paper still mentions "slope of 18%" without indicating whether this is a positive or a negative slope (i.e. terrain going up or down from antenna).  This is important for a clear understanding.  It only involves adding a sentence to clarify the sign convention.

3. Old comment, line 227: "range of footprint will be greatly stretched".  The author's answer is not convincing.  If I'm not mistaken, the reflection point will be closer to the antenna on tilted surfaces (ground going down).  It is unclear to me how this can cause the footprint to stretch when considering tracks covering a given satellite elevation interval (5 to 20 degrees).  This requires some explanation in the paper, possibly with a drawing showing the direct and reflected paths.

4. New comment, line 220: this must be updated to reflect that Figure 5 shows both satellites 7 and 8 (and not only 7).

New typos:

line 100: "The PBO site p025 located"  --> "is located"

line 364: "dues" --> "due"
